# Chemistry of Tropical Eucheumatoids: Potential for Food and Feed Applications

**DOI:** 10.3390/biom11060804

**Published:** 2021-05-29

**Authors:** Andrea Ariano, Nadia Musco, Lorella Severino, Anna De Maio, Annabella Tramice, Giuseppina Tommonaro, Sara Damiano, Angelo Genovese, Oladokun Sulaiman Olanrewaju, Fulvia Bovera, Giulia Guerriero

**Affiliations:** 1Department of Veterinary Medicine and Animal Production, University of Napoli Federico II, 80100 Napoli, Italy; andrea.ariano@unina.it (A.A.); nadia.musco@unina.it (N.M.); lorella.severino@unina.it (L.S.); sara.damiano@unina.it (S.D.); fulvia.bovera@unina.it (F.B.); 2Department of Biology, University of Napoli Federico II, 80126 Napoli, Italy; angelo.genovese@unina.it (A.G.); giulia.guerriero@unina.it (G.G.); 3National Research Council-Institute of Biomolecular Chemistry, 80078 Pozzuoli, Italy; atramice@icb.cnr.it (A.T.); gtommonaro@icb.cnr.it (G.T.); 4School of Ocean Engineering, Universiti Malaysia Terengganu, Kuala Terengganu 20130, Malaysia; oolanrewaju8@gmail.com; 5Interdepartmental Research Centre for Environment, University of Naples Federico II, 80134 Naples, Italy

**Keywords:** *Kappaphycus*, feed additive, heavy metals, trace elements, antioxidants

## Abstract

The use of seaweeds as additives in animal nutrition may be a valid option to traditional feed as they represent a rich source of minerals, carbohydrates and antioxidants. The aim of this study was to analyze the chemical composition and in vitro antioxidant capacity of two tropical eucheumatoids, *Kappaphycus alvarezii* and *Kappaphycus striatus*, in Malaysian wild offshore waters. The chemical analysis was performed via inductively coupled plasma–optical emission spectroscopy for evaluating the concentration of toxic (Cd, Pb, Hg, As) and essential elements (Mn, Fe, Cu, Ni, Zn, Se); NMR spectroscopy was used for carrageenans investigation. Furthermore, the soluble and fat-soluble antioxidant capacities were determined by FRAP, DPPH and ABTS assays. The chemical analysis revealed a higher content of trace elements in *K. alvarezii* as compared to *K. striatus*, and both exhibited a high mineral content. No significant differences in metal concentrations were found between the two species. Both samples showed a mixture of prevailing κ- and t-carrageenans. Finally, the levels of soluble and fat-soluble antioxidants in *K. alvarezii* were significantly higher than in *K. striatus*. Our findings suggest that *K. alvarezii* could be used as a potential feed additive because of its favorable chemical and nutritional features.

## 1. Introduction

Currently, the increased interest in the use of natural ingredients in animal nutrition in order to improve health is resulting in the rediscovery of ancient ingredients. Seaweeds have been used to feed livestock for thousands of years and have been mentioned in Ancient Greece and in the Icelandic sagas [1]. There are many species of seaweeds with different properties: some of them are rich in protein and can thus be used as alternative protein sources in animal nutrition, while others are mainly sources of bioactive compounds [2]. In particular, since the early 2000s, seaweed supplementation has been known for the prebiotic action of their complex carbohydrates. The inclusion of seaweeds in the feed ration at low levels can exert a potent prebiotic activity in the gastrointestinal tract, improving stress resistance, immune system competency and productivity, and reducing pathogen load. Thus, animal health and resistance to disease can increase, as well as performances, similarly to what happens with the inclusion of antibiotics at sub therapeutic levels for growth promotion purposes, but without the risk of developing antibiotic resistance [3].

Red seaweeds are a critical source of numerous bioactive compounds, in contrast to the green and brown seaweeds which are a source of polysaccharides such as sulfated galactans (carrageenans or agars), minerals, unsaturated fatty acids, amino acids, vitamins, phycobiliproteins, other pigments, phycolectins and mycosporine-like amino acids, which have many biological and industrial applications [4].

Among the red seaweeds, *Kappaphycus alvarezii* (Doty) L. M. Liao and *Kappaphycusstriatus* (F. Schmitz) L. M. Liao are two of the most widely cultivated *Kappaphycus* species and are extensively cultivated as a source of carrageenan. They have been used for many years in food applications and are generally thought to be safe. In fact, they present thickening, gelling, emulsifying and stabilizing abilities while being indigestible and having no nutritional value [5].

Carrageenans are also employed in several nondairy food products (instant products, jellies, pet foods, sauces) and nonfood products (pharmaceutical formulations, cosmetics and oil well drilling fluid) [4]. They are also widely used as an inflammatory inducer in experimental animal models of inflammation for research and drug development purposes [6].

In the pharmaceutical industry, carrageenans have attracted interest as useful excipients for sustained-release formulations and other drug delivery systems [7].

From the chemical point of view, these molecules are a skeleton made of alternating 3-linked β-D-galactopyranose (G-units) and 4-linked α-D-galactopyranose (D-units) or 4-linked 3,6-anhydro-α-D-galactopyranose (DA-units), forming the α disaccharide repeating unit of carrageenans [8]. The different polysaccharidic carrageenan structures are related to the presence of the sulfate groups and their variable positions within the disaccharide units.

The three commercially most important carrageenans are called ι-, κ- and λ-carrageenan, and the corresponding IUPAC-related names and letter codes are carrageenose 2,4’-disulfate (G4S-DA2S), carrageenose 4’-sulfate (G4S-DA) and carrageenan 2,6,2’-trisulfate (G2SD2S, 6S).

Seaweeds have the ability to accumulate large amounts of minerals from their environment. This is often considered a positive characteristic for the essential elements but a negative characteristic for the toxic elements. High levels of toxic elements are an issue that limits their use [9,10] in observance of EU regulation 1881/2006 on acceptable limits of heavy metals. Therefore, seaweeds may represent a mineral additive for animal diets, but the content of heavy metals may induce toxicological effects on the organism [11].

Seaweeds are also considered an important source of antioxidant substances able to protect the animal against reactive oxygen species [12,13,14].

It is known that although seaweeds are exposed to free radical and strong oxidizing agents due to the reaction between sunlight and oxygen, their structural component does not show any oxidative damage [15]. This evidence has allowed the hypothesis that seaweeds, as all living organisms, are capable of generating essential defense mechanisms, including antioxidants, as well as efficient secondary metabolites against the oxidation [16,17,18].

There are several studies in the literature on the chemical-nutritional characteristics and benefits of the utilization of seaweeds in animal nutrition [19,20], but only a few studies have focused on the identification of the toxicological risks associated with their use [10,11].

Seaweed culture in Malaysia started in 1978, as introduced by the well-known U.S. company Aquatic Resources Limited. Since 1980, the Department of Fisheries has managed these activities with the support of government authorities and universities that are striving to improve the seaweed industry. Cultured seaweeds have been identified as one of the highest-value commodities under the National Key Economic Area for Agriculture (NKEA) Entry Point 3 (PPE3) project and many other Malaysian National Agri-food Policy projects (2011–2020) (NAP 4) [21].

Seaweeds are a valuable alternative ingredient for livestock feeds, and algae, in particular, have been identified as an important protein source for the future [22]. Nutritionists are working to find alternative protein sources that can replace traditional sources such as soybeans [23]; thus, algae can be an option. In addition, seaweeds are a rich source of bioactive compounds such as natural pigments, carotenoids and polyunsaturated fatty acids that can improve the quality of animal products [24]. On 1 January 2006, the use of antibiotics as growth promoters was forbidden by the World Health Organization. This ban forced industry professionals to search for alternative natural solutions [25]. Thanks to the prebiotic effects of algae oligosaccharides, algae can be part of the solution [24].

In light of these considerations, the aim of this study was to evaluate the chemical composition, including the presence of carrageenans and the concentration of toxic (Cd, Pb, Hg and As) and essential elements (Mn, Fe, Cu, Ni, Zn and Se), and the total soluble and fat-soluble antioxidant capacities in two eucheumatoid seaweeds of the genus *Kappaphycus* Doty (Gigartinales, Rhodophyta), *K. alvarezii* and *K. striatus* collected wild offshore of Palau Bidong (Malaysia) to define their potential use as an additive in animal nutrition.

## 2. Materials and Methods

The experiment took place offshore of Palau Bidong Island, which is one square kilometer in area and accessible from the coastal town of Merang, located at 05°36 N and 103°03 E. Seaweeds were planted and collected in the wild during August 2018 and repeated in August 2019 using identical procedures as follows:

### 2.1. Biological Material

After preliminary morphological identification, as described in Doty [26] and also as shown by Li et al. [27], samples of red seaweeds *K. alvarezii* and *K. striatus* were planted in baskets with dimensions of 26(L) × 19 (W) × 9 (H) cm, and a volume of 4.45 L, covered with fishing net (1.5 cm mesh size) (Figure 1). Two lines of three baskets each were used for a total of 6 baskets per species.

After 14 days in August 2018 and again after 14 days in August 2019, the seaweeds were collected in a manner as reported by Olanrewaju et al. [28]. Averaging the measurements from both years, the mean ± SD initial total weights, when planted, were 22.3 ± 0.3 g (*n* = 12) for *K. alvarezii* and 25.2 ± 0.5 g (*n* = 12) for *K. striatus.* After the 14 day experimental periods, the mean ± SD final weights were 3173 ± 120 and 3239 ± 103 g for *K. alvarezii* and *K. striatus,* respectively, and the daily growth rates (DGRs) were 5.30% and 5.45% per day with corresponding biomasses of 225 ± 18 and 231 ± 15 g for *K. alvarezii* and *K. striatus* respectively.

After sampling, the seaweeds were rinsed with milli-Q water until arrival at the laboratory. Then, seaweeds grown in the six different baskets/species were dried (Figure 2) for 48 h in a ventilated oven at 50 °C (TCF50, Sigma Precision S.r.l., Dese, Italy) and stored at room temperature in screw cap jars (PL22, Lamaplast, Sesto San Giovanni, Italy), maintaining the separation between the contents of one basket and the other. All chemical and biochemical analyses were carried out on two separate aliquots of dried specimens taken from each of the six baskets. Each analysis was performed in triplicate, resulting in 36 analyses per species per year. As the 2018 and 2019 analyses yielded very similar results, they were combined for an aggregate of 72 total analyses per species to be used for the final statistical analysis.

### 2.2. Mineral and Heavy Metal Analysis of K. alvarezii and K. striatus

The dried seaweed samples (0.5 g) were ground to pass through a 1 mm screen (Brabender Wiley mill, Brabender OHG, Duisburg, Germany). Subsequently, the chemical composition of seaweed samples was determined as follows: nitrogen-free extracts (NFE, as % of dry matter), dry matter (DM), ash, crude protein (CP), ether extract (EE), and crude fiber (CF) (procedure numbers 934.01, 942.05, 954.01, 920.39, and 962.09, respectively, according to AOAC) [29]. For trace elements determination, glassware and laboratory equipment were decontaminated before use with diluted ultrapure 65% HNO_3_ (RomilUpA, Cambridge, UK) and were rinsed with Milli-Q water (Millipore Corp., Bedford, MA).

Before inductively coupled plasma–optical emission spectroscopy (ICP-OES) analysis, the sample was placed in a Teflon vessel with 5.0 mL of 65% HNO_3_ and 2.0 mL of 30% H_2_O_2_ (RomilUpA). The vessel was sealed and placed in a microwave digestion system (Milestone, Bergamo, Italy). Microwave-assisted digestion was performed with a mineralization program for 25 min at 200 °C, as reported by Ariano et al. [30].

The concentration of trace elements was determined via the ICP-OES technique, employing a Perkin Elmer Optima 2100 DV instrument with a CETAC U5000AT. Calibration curves on the sample and on two blanks were run during each set of analyses to test the purity of the chemicals. Reference materials (CRM DORM-4, NRC, Canada) were also included for internal control. All the values of the reference materials were within certified limits reported by the ISO/Guide 30:2015. Instrumental detection limits are expressed as wet weight (w.w.) and determined following the protocol described by Barnard et al. [31].

### 2.3. Extraction and NMR Spectroscopy Analysis of Carrageenan

After sampling, algae were dried and stored until analysis. For the chemical analysis of carrageenans, 10 g of each dried sample was extracted according to the standard procedure with distilled water. All extracts were filtered and evaporated under vacuum. Aqueous extracts were dissolved in hot distilled water and precipitated with cold ethanol with the aim of isolating the polysaccharidic fraction containing the carrageenan molecules. These were recovered after centrifugation at 10,000 rpm for 40 min, and they were lyophilized and then analyzed by NMR spectroscopy. NMR samples were prepared by dissolving the sonicated carrageenan fractions (20 mg mL^−1^) in D_2_O (0.650 mL) containing 0.05% of 3-(trimethylsilyl)-propionic-2,2,3,3-d4 acid and sodium salt as internal standard (TSP). ^1^H and ^13^C NMR spectra were taken at 65 °C on a Bruker Avance 400 spectrometer equipped with a Cryo-Probe Prodigy operating at 400.132 MHz. Typically, experiments were acquired with an interpulse delay of 5 s (D1) and 13,000 scans for each experiment. Chemical shifts were referenced to the internal TSP standard (−0.017 ppm for ^1^H and −0.18 for ^13^C signals, respectively).

### 2.4. Extraction of Soluble and Fat-Soluble Antioxidants and Determination of Antioxidant Capacity

The extraction of soluble and fat-soluble antioxidants was conducted by making some modifications to the procedure previously described [32]. Of note, 0.3 g of driedseaweeds were sonicated for 20 min and subsequently homogenized by a Polytron Ultra Turrax T8 (IKA-WERKE). The extraction of the water-soluble antioxidants was carried out in the dark for 2 h in a mixture consisting of methanol, water and formic acid (80:20:0.1 by vol). Subsequently, extracts were centrifuged at 3500 rpm for 1 min, at 4 °C, using an Eppendorf 5417 R centrifuge (Bio-Rad, rotor F 45-30-11). The obtained supernatant (S1), which represents the hydrophilic extract, was stored at 4 °C, while the pellet was subjected to two subsequent extractions. The hydrophilic extracts S1, S2 and S3 were combined and further centrifuged at 10,000 rpm for 5 min at 4 °C. The extraction of fat-soluble antioxidants was carried out in acetone by performing the same experimental procedure described above.

The soluble and fat-soluble extracts were directly used to measure the total antioxidant capacity by ABTS, FRAP and DPPH assays using five different samples each year, and the tests were carried out in triplicate. The ABTS assay measured the free radical scavenging activity via 2,2′-azino-bis(3-ethylbenzthiazoline-6-sulphonic acid) (ABTS^•+^) radical cation decolorization [33]. The ABTS^•+^ radical cation was obtained by a reaction between 7 mM of ABTS and 2.45 mM of potassium persulfate. The reaction mixture was left for 16 h in the dark at room temperature and used within two days. The ABTS^•+^ solution was diluted with ethanol, to produce an absorbance of 0.700 ± 0.050 at 734 nm. Fifty microliters of samples were mixed with 1.9 mL of diluted ABTS^•+^ solution. After 6 min at room temperature, the absorbance of the samples was measured at 734 nm. Trolox solution (final concentration of 0–15 µM) was used as a reference standard. The results were expressed as µmol Trolox g^−1^ fresh weight of tissue and were calculated as the mean value ± standard deviation (*n* = 36 for each year). A FRAP assay measured the reducing potential of an antioxidant reacting with a ferric tripyridyltriazine (Fe^3+^-TPTZ) complex and producing a colored ferrous tripyridyltriazine (Fe^2+^-TPTZ). The antioxidant capacity of examined samples was determined spectrophotometrically, as described in Benzie and Strain [34]. The method is based on the reduction of the Fe^3+^ TPTZ complex (colorless complex) to Fe^2+^-tripyridyltriazine (blue-colored complex) formed by the action of electron-donating antioxidants at low pH. The Ferric reducing antioxidant power (FRAP) reagent was prepared by mixing 300 mM of acetate buffer and 10 mL of TPTZ in 40 mM of HCl and 20 mM of FeCl_3_∙6H_2_O in the proportion of 10:1:1 at 37 °C. The FRAP reagent (3.995 mL) was mixed with 5 μL of sample. An intense blue color complex was formed when the ferric tripyridyltriazine (Fe^3+^-TPTZ) complex was reduced to the ferrous (Fe^2+^) form. The absorbance at 593 nm was recorded against a reagent blank containing 3.995 mL of FRAP reagent and 5 μL of distilled water after 30 min of incubation at 37 °C. The calibration curve was obtained by plotting the absorbance at 593 nm versus different concentrations of FeSO_4_. The concentrations of FeSO_4_ were, in turn, plotted against the concentration of standard antioxidant Trolox. The FRAP values were obtained by comparing the absorbance change in the test mixture with those obtained from increasing concentrations of Fe^3+^. The values were expressed as micromoles of Trolox per gram of sample and calculated as the mean value ± standard deviation (*n* = 36 for each year). Finally, the total antioxidant capacity was also measured by the DPPH method. A solution of 60 µM of 1,1-diphenyl-2-picrylhydrazyl (DPPH^•^) in ethanol was prepared daily in the dark. The absorbance of the solution was adjusted to 0.650 ± 0.050 at 517 nm using fresh ethanol. Then, 50 µL of standard or sample was mixed with 1.95 mL of DPPH^•^ solution and incubated for 15 min in the dark. The decrease in absorbance was monitored at 517 nm at room temperature. The control consisted of 50 µL of ethanol in 1.95 mL of DPPH solution. The standard curve was obtained by measuring DPPH^•^ scavenging activities of 6.25, 12.5, 18.8 and 25 µM of Trolox. The results were expressed as µmol Trolox g^−1^ fresh weight of tissue and were calculated as the mean value ± standard deviation (*n* = 36 for each year).

### 2.5. Quality Assurance

The performance of the method was assessed through participation in inter-laboratory studies organized by FAPAS (Food Analysis Performance Assessment Scheme, Sand Hutton, UK).

### 2.6. Statistical Analysis

The results related to the determination of chemical characteristics, mineral and heavy metal composition, biomass, and total soluble and fat-soluble antioxidant capacities obtained in 2018 and 2019 were very similar; thus, they were reported as the mean ± standard deviation (SD) of the combined results for the two years. Differences in chemical characteristics, mineral and heavy metal composition, and biomass results between the two species, *K. alvarezii* and *K. striatus,* were analyzed using one-way ANOVA. The root-mean-square error (RMSE) for these variables was also reported. Total soluble and fat-soluble antioxidant capacities measured via FRAP, DPPH and ABTS tests were analyzed by the Mann–Whitney U test and they are shown as mean ± SD. Statistical parameters are reported in figures and legends. Differences were considered significant when *p* < 0.05.

## 3. Results

### 3.1. Mineral and Heavy Metal Composition of K. alvarezii and K. striatus

Table 1 shows the mineral and heavy metal composition of the two seaweeds studied in this trial. *K. alvarezii* showed a significantly higher percentage of dry matter and ash, and a significantly lower percentage of nitrogen-free extracts (NFE) than *K. striatus*; *K. alvarezii* showed a significantly higher amount of Al, Fe and Mn in comparison to the other seaweed.

No significant differences were found between metal concentrations, with the exception of the As, which was significantly higher in *K. alvarezii*.

### 3.2. Characterization of K. alvarezii and K. striatus Carrageenans

^13^C- and ^1^H-NMR spectra of polysaccharidic samples from *K. alvarezii* and *K. striatus* were recorded and investigated. Data are shown in Figure 3; anomeric carbon signals and α-anomeric protons of monosaccharide units were reported for both samples. In the ^13^C spectra, the chemical shift values of the most intense signals suggested the presence of α-DA monosaccharide unit, typical of κ-carrageenan and α-DA2S monosaccharide unit, typical of ι-carrageenan.

In the ^1^H-NMR spectra, the α-anomeric signals α-DA and α-DA2S units were integrated, and taking 1 as a reference value, the integral of the signal of TSP standard and their values were compared. In particular, if, in the *K. alvarezii* fraction, the anomeric protons ratio between α-DA (4.89 ppm; ^13^C 97.80 ppm) and α-DA2S (5.28 ppm; ^13^C 94.905 ppm) integrals was 1.46, in the case of the *K. striatus* fraction, the anomeric protons ratio between α-DA (4.84 ppm; ^13^C 97.87 ppm) and α-DA2S (5.27 ppm; ^13^C 94.901 ppm) integrals would be 2.15. In this last case, although κ-carrageenan polysaccharide was most abundant, a significant signal at 5.42 ppm in the ^1^H-NMR spectra also suggested the presence of -and λ-carrageenans.

### 3.3. Soluble and Fat-Soluble Antioxidant Capacity of K. alvarezii and K. striatus

Total soluble and fat-soluble antioxidant activities were determined using FRAP, DPPH and ABTS assays to check whether there are significant differences between the data obtained by these three spectrophotometric assays and to identify among them which is most suitable for measuring the antioxidant capacities in *K. alvarezii* and *K. striatus*.

In both species, the soluble and fat-soluble antioxidant activities measured with the FRAP test were always lower than those measured with the other two assays. Moreover, no significant differences occurred, when the soluble and fat-soluble antioxidant capacities were determined by the DPPH and ABTS test (Figure 4a,b, respectively).

Nonetheless, the values of soluble and fat-soluble antioxidants measured in *K. alvarezii* were always higher than those determined in *K. striatus*, regardless of the type of assay performed (Figure 5a,b, respectively).

## 4. Discussion

Seaweeds are acquiring considerable importance in the biotechnological, environmental and nutraceutical fields. The wide phenotypic and ecological diversity among seaweeds suggests that they possess a wide range of functional capabilities encoded in their genomes [35]. Moreover, it is well known that their global demand as a food source is growing. Much research has shown that food products derived from duckweed [17] and seaweeds have substantial benefits for human health [21].

The trend of increasing nutritional demand for seaweed products derives not only from a greater attention towards health but also from a wider use as food additives. In addition to their nutritional value, seaweeds are increasingly being commercialized as “functional foods” or “nutraceuticals”; these terms have no legal status in many countries but they describe foods that contain bioactive compounds or phytochemicals, which can improve health (e.g., as anti-inflammatories, and for disease prevention) [36,37].

Although the nutritional potential or bioactive content of different seaweeds has been amply demonstrated, there are still few studies conducted to quantify the bio-availability of nutrients and phytochemicals from seaweed foods, and very minimal data are available on the identification of toxicological risks associated with their use [9,38].

In light of this knowledge, we have evaluated the main chemical constituents, the antioxidant power and the total concentrations of Cd, Pb, Hg and As in *K. alvarezii* and *K. striatus* collected in Palau Bidong, Malaysia in order to define their potential use as additives in animal nutrition and to assess the health risk associated with animal consumption.

There were several reasons for which the *Kappaphycus* genus was selected to be used in this research. The two species of *Kappaphycus* that are commonly cultured in South east Asia are *Kappaphycus striatus* and *Kappaphycus alvarezii* [4,21]. A field observation and cultivation carried out for many years by Olanrewajuet et al. [28] highlighted that *K. alvarezii* and *K. striatus* perform very wellwild offshore of Palau Bidong in intertidal zones where they are exposed to fluctuating environmental conditions and at reported physical-chemical sea water parameters. Further, we can assume that the effects of abiotic factors on both cultivated *K. alvarezii* and *K. striatus* were similar because they were located in the same wild offshore location.

We collected both species on the 14th day after planting when the growth rate was at its maximum. From day 14 to day 28, the growth rates for both *Kappaphycus* species decreased with these possible explanations: nutrient availability; light and space competition between seaweeds; and tropical sea surface temperature (26.8 °C),which slows growth synergy due to the high metabolism rate.

Our results on the relative abundance of trace elements in the cultured field show differences in their accumulation between the two seaweeds. The full amount of trace elements detected in *K. alvarezii* (42.75 mg k^−1^) was about double that of *K. striatus* (22.85 mg k^−1^) in keeping with the differences found in ash levels. 

Further, both seaweeds have high minerals contents. This may be due to their cell wall polysaccharides and proteins with anionic carboxyl, sulphate and phosphate groups as excellent binding sites for metal retention [39]. The bio-absorption capacities of seaweeds have been shown to be significantly affected by several environmental factors, such as water temperature and pH [38]. Nevertheless, the concentration of minerals within the environment do not always reflect their bioavailability. It is well known that mineral bioavailability is influenced by solubility, interaction between nutrients, circadian differences, gastric properties, metabolic differences, disease, and many other parameters, as well as families, genera and species of seaweed [40,41]. Additionally, the bio-availability of heavy metals for seaweeds varies consistently with their geographical origin and harvesting time, and this might be responsible for the differences in our findings in terms of their relative abundance.

Interestingly, the higher levels of Fe found in our studies were also supported by the results reported by Chuah and Teo [42] who found Fe as the most represented heavy metal in *K. alvarezii* (14.9 ppm). From a nutritional point of view, the abundances of Fe, Cu, Mn, Se and Zn found in the two seaweeds in the present trial are exceedingly important as trace elements are obtained by animals via the dietary intake of feeds and are essential for health and immunity [43,44], growth [45,46], production [47,48] and reproduction [43,49]. The amount of Ni available in seaweeds should not be overlooked when the ingredients are included in diet formulation, considering the full amount supplied by the other ingredients. However, a recent opinion by the European Food Safety Authority, EFSA [50], confirmed that adverse effects from Ni in the feed are unlikely to occur in numerous species [51]. Regarding Al, it is a ubiquitous metal and is rarely toxic when accumulated within the animal organism, although some adverse effects of Al have been observed on the nervous and reproductive systems of animals. However, the EFSA [52] has established a tolerable weekly intake of one milligram of Al per kilogram of bodyweight.

Compared to maximum levels of heavy metals set by the EU Commission, Cd, Pb, As and Hg levels detected in seaweeds in this study were always less than the maximum permissible levels of these elements established for feed materials. In fact, the EU regulation establishes the maximum permissible levels of Cd, Pb, As and Hg in feed materials as follows: 1.0 mg kg^−1^ for feed materials of vegetable origin, 10.0 mg kg^−1^ for seaweed meal and 40.0 mg kg^−1^ for feed materials derived from seaweed [53].

Overall, our results indicate that the danger of exposure to heavy metals from the consumption of seaweeds is comparatively low and in compliance with EU regulations. Further studies on a greater number of samples are needed for metals and other pollutants, but concerning Pb, Cd, As and Hg concentrations, these preliminary results support the possible use of these seaweed species for animal feeding with no additional hazards.

Comparing the characteristics of *Kappaphycus* spp. with other seaweeds, some similarities emerged. Roleda et al. [54] analyzed various chemical constituents associated with both health benefits (carbohydrates, protein, fatty acids and minerals) and health risks (heavy metals) in *Ulva lactuca* (Chlorophyta) strains. Similar to our results, this seaweed showed low protein and lipid contents, similar to other vegetable sources (i.e., celeriac, celery, Brussels sprouts, and broccoli). In addition, the *Ulva lactuca* microelements and macroelements concentrations were sufficient to contribute to the daily dietary mineral intake. The levels of heavy metals (As, Cd, Hg and Pb) were found at low levels, without representing a health risk. Other authors investigated the composition of three seaweeds: the brown algae *Saccharina latissima* and *Alaria esculenta*, and the red alga *Palmaria palmate* [55]. From these findings, all three species emerged as good sources of antioxidants, and the heavy metal concentrations were below the upper limits set by the EU Commission Regulation on contaminants in foodstuffs. In order to assess the use of seaweeds as ingredients of ruminant diets, de la Moneda et al. [56] analyzed the chemical composition and in vitro rumen fermentation of eight seaweed species (brown, red and green) collected in Norway during spring and autumn. Interestingly, many differences emerged among different seasons. In particular, the degradability after 24 h of fermentation was greater in spring than in autumn, with *Palmaria palmata* showing the greatest value and *Pelvetia canaliculata* the lowest. Seaweeds are different for their fermentation pattern, and autumn *Alaria esculenta*, *Laminaria digitata*, *Saccharina latissima* and *Palmaria palmate* had fermentation patterns similar to those of high-starch feeds [55,56]. However, the inclusion of all the seaweeds in the diet (up to 200 g/kg of concentrate) produced only slight effects on in vitro ruminal fermentation and can be used as an additive in the feed ration.

The nutritional value attributed to macroalgae makes them particularly appropriate for use in animal feeding. Surely, their potential use in fish farming seems to be the more interesting one, and there is a great quantity of data that support this use. The advantages range from improved growth and development rates, disease resistance, financial gain and even ecological preservation [57,58,59,60].

In poultry diets, it appears to be possible to enrich broiler feed with green seaweed, or a mixture of green and red seaweed, in order to stimulate both the growth and the health of the broilers [61,62,63]. 

Interestingly, various seaweed species (either green, red or brown) have the potential to enhance various qualities of poultry eggs (in terms of weight, yolk cholesterol reduction and, depending on the species, other bioactive molecules capable of even improving the microflora of the digestive system of poultries). It appears that a mixture of brown, green and red seaweed could be a promising supplement used in order to enrich eggs [64,65,66].

In ruminant feeds, the use of seaweed has been affected by the high demand for animal feed protein, the need for alternatives to the traditional soybean and animal protein feed, as well as the food market regulations related to livestock feeding methods. Studies carried out to date regarding the use of seaweed in bovine, caprine and other ruminant nutrition have focused on the addition of small quantities of different macroalgal species to the feed and the subsequent assessment of the animal to check for possible prebiotic activity, reduced methane emission and enhanced animal performance. Considering the data in the literature, algae seem to be useful for these purposes [67].

Seaweed can be included in the diet as a pellet binder in poultry diets at up to 5–15% of the diet. An inclusion up to 3% of the diet improves the hardness of the pellet [68]. With duck diets, brown seaweeds can be included in the starter and finisher diets up to 12% and 15%, respectively, without adversely affecting growth performance and meat quality. Red seaweed (*Polysiphonia* spp.) can be included at up to 15% in duck starter and grower diets with no adverse effects on growth performance or carcass quality [69]. However, some studies have indicated that including as little as 10% seaweed in a broiler diet reduced growth performance [70]. Recently, Morais et al. [71] stated that seaweeds demonstrated a potential to be further explored as an animal feed additive/supplement but cannot be applied as a complete substitute of the typical animal feed. Seaweed beneficial effects are observed at generally below 10% of the total concentration in the animal feed; above that, it was demonstrated to show negative effects, and animals even refused to eat the provided feed.

Commercial seaweed varieties that are farmed for carrageenan production in Malaysia belong to the genera *Kappaphycus* and *Eucheuma* [4]. Cultivation is spreading into subtropical regions. Currently, the supply of carrageenan is not adequate for the global demand. The sea area available for farming is one limiting factor in the production of seaweeds for carrageenan extraction [72]. The most important types of commercial carrageenans are κ-, ι- and λ-carrageenan. κ-carrageenans occur in the cell wall of some species of marine red algae, such as *Chondrus* sp., *Gigartina* sp. and *Eucheuma* sp., but are mostly extracted from tropical seaweeds such as *K. alvarezii* [73], while ι-carrageenans are mainly extracted from *Eucheuma spinosum*. λ-carrageenans are extracted from red algaewithin the *Gigartina* and *Chondrus* genera, which produces this type of polysaccharide during the sporophytic stage [7].

The spectroscopic characterization of polysaccharidic extracts of *K. alvarezii* and *K. striatus* collected at Palau Bidong suggested, for both species, a mixture of prevailing κ- and ι-carrageenans, in agreement with data previously reported in literature for other red seaweed species [74]. In *K. alvarezii* fraction the anomeric protons ratio between α-DA and α-DA2S integrals was of 1.46. In the case of *K. striatus* fraction, the anomeric protons ratio between α-DA and α-DA2S integrals was of 2.15. In this latter case, although κ-carrageenan polysaccharide was the most abundant, a significant signal at 5.42 ppm in ^1^H-NMR spectra suggested the presence also of ι- and λ-carrageenans.

These data presented interesting results considering that these polysaccharides display physicochemical properties (thickening, gelling, emulsifying and stabilizing abilities) that make them useful as components of many food products. Examples include their use in cottage cheese, puddings and dairy desserts to improve their texture and as binders and stabilizers in the meat processing industry for the production of hamburgers and sausages [7].

Furthermore, the levels of total soluble and fat-soluble antioxidant activity measured by FRAP, DPPH and ABTS assays are always higher in *K. alvarezii* compared to those determined in *K. striatus*, suggesting that this species is very interesting in terms of nutrition. ABTS and DPPH tests seem to be more suitable than the FRAP assay for these determinations, as the antioxidant activity levels via FRAP test are always lower than those measured by ABTS and DPPH assays. These results may be attributed to the fact that the FRAP test cannot detect compounds that act by radical quenching (hydrogen transfer); particularly, it does not allow the determination of thiols (as glutathione) [75].The high content of soluble antioxidants may provide intracellular and extracellular aqueous phase antioxidant capacities primarily by scavenging oxygen free radicals [76,77]. Furthermore, its high fat soluble concentration, instead, should represent a defense against oxidant-induced membrane injury [15,17] and contrast many pathological conditions and diseases [78,79] inhibiting or neutralizing the effects of free radicals.

These properties could contribute to improving animal welfare and sustainability, and to reducing the use of antibiotics. In addition, the use as a nutrient supplement has the advantage of requiring only a small amount of seaweeds to be added to the feeds, therefore reducing the risks related to the accumulation of toxic elements. 

## 5. Conclusions

Based on the chemical-nutritional characteristics, the content of essential elements and the negligible levels of heavy metals determined in this research, it can be concluded that the two tropical eucheumatoids, *Kappaphycus alvarezii* and *Kappaphycus striatus*, planted in Malaysian wild offshore waters could be used in animal nutrition, but, because of the low amount of protein and fat, as a percentage of its dry matter, its best use could be as a mineral additive in animal feed under intensive production. Its properties may contribute to improving the immune response due to the prebiotic action of its complex carbohydrates, which may result in a reduction in the use of antibiotics. 

In addition, *K. alvarezii* could be marketed as “functional foods” or “nutraceuticals”; as its high ash content and soluble and fat-soluble antioxidant capacity may benefit health beyond the role of basic nutrition. In fact, the antioxidants introduced by this seaweed may enhance the physiological defense, neutralizing the toxic effects of free radicals. The high nitrogen-free extract could also exert a prebiotic effect in the gut, although further studies are needed in order to confirm this hypothesis. More research is required to determine how the application of *K. alvarezii* may be utilized to improve farm management practices in order to mitigate the possible health risks related to animal consumption.

## Figures and Tables

**Figure 1 biomolecules-11-00804-f001:**
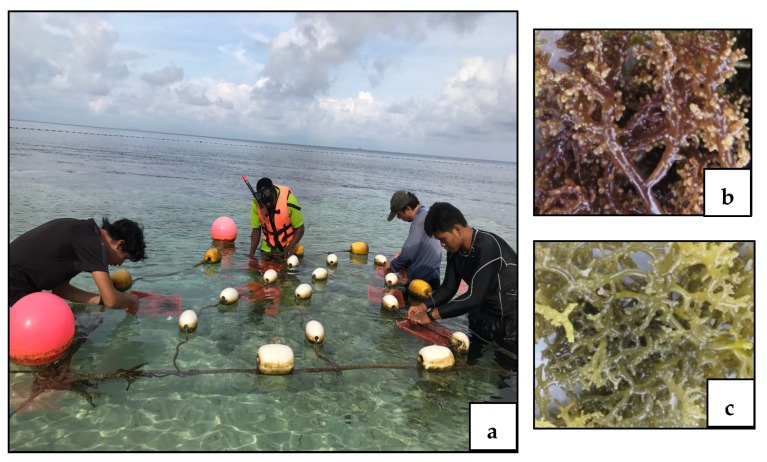
Basket system (**a**) of tropical eucheumatoids: *Kappaphycus alvarezii* (**b**) and *Kappaphycusstriatus* (**c**). Photo (**a**) was taken with farmer’s permission.

**Figure 2 biomolecules-11-00804-f002:**
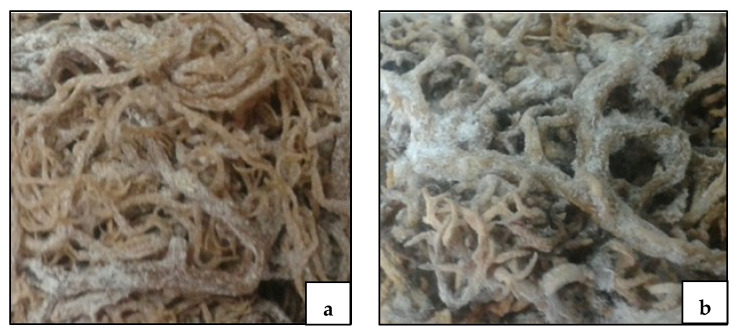
Dried samples of tropical eucheumatoids examined:(**a**) *Kappaphycus alvarezii*; (**b**) *Kappaphycus striatus*.

**Figure 3 biomolecules-11-00804-f003:**
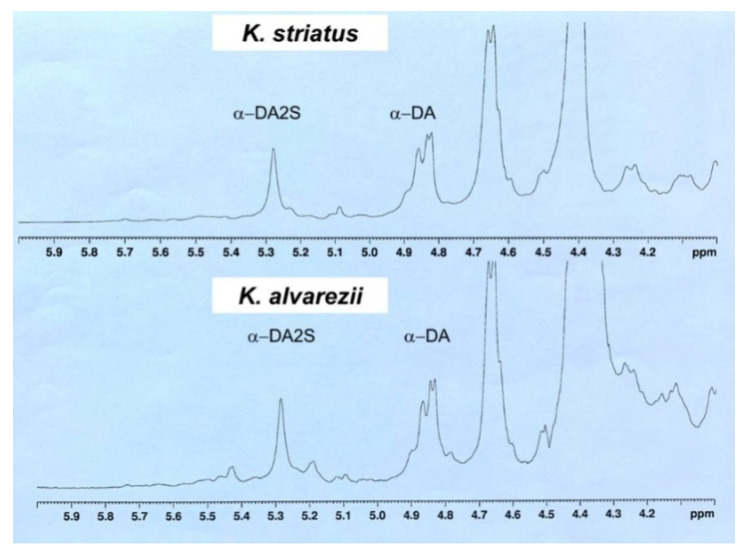
Anomeric region of the ^1^H-NMR spectra of the carrageenans from *K. alvarezii* and *K. striatus*.

**Figure 4 biomolecules-11-00804-f004:**
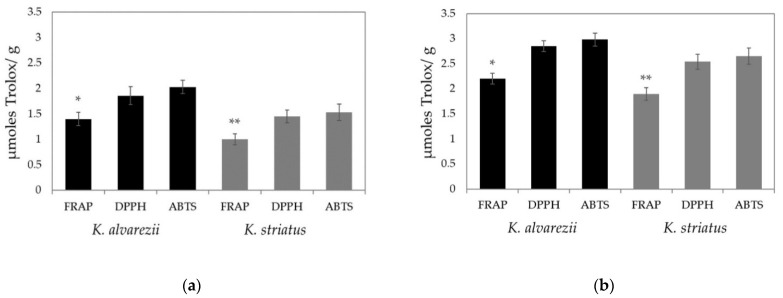
Total soluble (**a**) and fat-soluble (**b**) antioxidant capacities determined in *K. alvarezii* and *K. striatus*. The data are represented as the mean ± SD (*n* = 36 for each year). The results were analyzed via Mann–Whitney U test: * significant difference (*p* < 0.05) of the soluble and fat-soluble antioxidants obtained via FRAP test compared to those determined via DPPH and ABTS assays in *K. alvarezii*; ** significant difference (*p* < 0.05) of the soluble and fat-soluble antioxidants obtained via FRAP test compared to those determined via DPPH and ABTS assays in *K. striatus*. No significant difference between the soluble and fat-soluble antioxidant capacity measured by the ABTS and DPPH assays is observable in either species.

**Figure 5 biomolecules-11-00804-f005:**
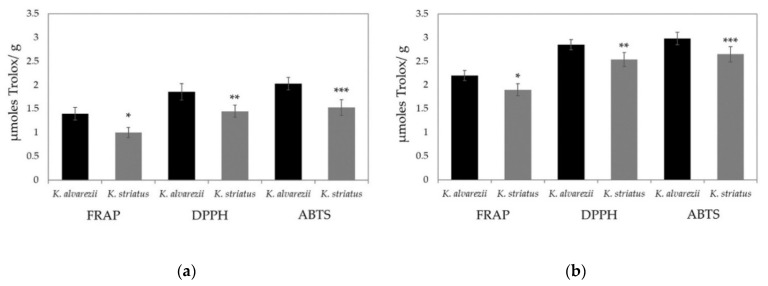
Total soluble (**a**) and fat-soluble antioxidant (**b**) capacities in *K. alvarezii* and *K. striatus* determined by FRAP, DPPH and ABTS assays. Data are represented as the mean ± SD (*n* = 36 for each year). Results were analyzed via Mann–Whitney U test: significant difference (*p* < 0.05) of soluble and fat-soluble antioxidant capacities obtained via FRAP (*), DPPH (**) and ABTS (***) test in *K. alvarezii* versus *K. striatus*.

**Table 1 biomolecules-11-00804-t001:** Chemical characteristics, mineral and heavy metal composition of dried *K. alvarezii* and *K. striatus*.

Chemical Composition (% Dry Matter)	*Kappaphycus* *alvarezii*	*Kappaphycus striatus*	RMSE	*p*-Value
Dry matter	88.93	83.88	1.87	0.0031
Ash	45.37	35.88	2.29	0.0088
Crude Protein	6.81	6.95	0.12	0.5216
Ether extract	0.38	0.47	0.084	0.1539
Crude fiber	4.36	5.67	0.28	0.0560
NFE	43.08	51.03	0.95	0.0029
Mineral composition (mg kg^−1^ dry matter)				
Al	0.915	0.307	0.262	0.0024
Cu	0.035	0.070	0.036	0.1177
Mn	0.091	0.070	0.013	0.0224
Fe	40.009	21.889	6.515	0.0070
Ni	0.033	0.033	0.0145	0.9991
Se	0.153	0.156	0.039	0.9076
Zn	0.265	0.216	0.084	0.3413
Heavy metal composition (mg kg^−1^ dry matter)				
As	0.097	0.082	0.010	0.0343
Cd	0.009	0.010	0.004	0.7319
Pb	0.004	0.004	0.002	0.9213
Hg	0.006	0.005	0.001	0.1992

Results are the mean of 36 analyses performed in each year, 2018–2019. NFE: nitrogen-free extracts; RMSE: root-mean-square error.

## Data Availability

Not applicable.

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
