# Peer review of "Chemistry of Tropical Eucheumatoids: Potential for Food and Feed Applications"

_biomolecules, 2021, doi:10.3390/biom11060804_

Round 1

Reviewer 1 Report

Ariano and co-workers conducted a comprehensive chemical characterization of the eucheumatoids Kappaphycus alvarezii and K. striatus. This is important contribution regardless if will be used for food (e.g., as salads, or condiment for different food applications) and/or feed applications. Why the authors focused on animal feed and nutrition is not well justified. Seaweed for feed applications will require huge biomass that will eventually be in competition with the carrageenan industry.

The authors should be able to review and include pertinent literatures on the use of seaweeds for food and feed. The authors should also be able to compare and contrast the chemistry between red, brown and green seaweeds. It is refreshing to see this paper reporting comprehensive chemistry of tropical eucheumatoids as most available studies looked into cold temperate species. Below are some literatures for citations on the food and feed applications of different seaweed species.

Whole manuscript, the English needs improvement. Manuscript for review has several typographical errors.

Change Title. Suggestion- Chemistry of tropical eucheumatoids: potentials for food and feed applications

Abstract needs improvement. Not comprehensive enough in describing the extent of chemical characterization done.

Line 28: Why K. alvarezii SHOULD be used in animal nutrition? Is this imperative?

Materials and method: require details.

  1. Are the seaweed samples wild or cultivated? How were the different species recognized from each other? Based on morphology (be categorical. Include pictures of the seaweed as a Figure)? They could look the same among cultivated/farmed specimen. See Roleda et al. 2021 Algae https://doi.org/10.4490/algae.2021.36.2.18
  2. How was the biomass sampling/collection done? What does n=6 pools/year mean? What is a “pool”? How much sample i.e. wet weight per replicate was collected. How were they dried for eventual chemical assays? Does this means that both species were collected 2 times (2018 and 2019)? Does all data represent mean of 2 years? This has implication on the statistical analysis. Comparison (1-way ANOVA) was only between species.
  3. Physico-chemical parameters of seawater. How was sampling done? Replications? Snapshot measurement? What is their relevance to the chemistry of the seaweed tissue/biomass?

Discussion: compare and contrast the chemistry of the 2 Kappaphycus species with other seaweeds with known food and feed applications (see recommended literatures below).

Roleda MY, Lage S, Aluwini DF, Rebours C, Brurberg MB, Nitschke U, Gentili FG (2021) Chemical profiling of the Arctic sea lettuce Ulva lactuca (Chlorophyta) mass-cultivated on land under controlled conditions for food applications. Food Chemistry 341: 127999. https://doi.org/10.1016/j.foodchem.2020.127999

Roleda MY, Marfaing H, Desnica N, Jónsdóttir R, Skjermo J, Rebours C, Nitschke U (2019) Variations in polyphenol and heavy meal contents of wild-harvested and cultivated seaweed bulk biomass: Health risk assessment and implication for food applications. Food Control 95:121–134. https://doi.org/10.1016/j.foodcont.2018.07.031

Roleda MY, Skjermo J, Marfaing H, Jónsdóttir R, Rebours C, Gietl A, Stengel DB, Nitschke U (2018) Iodine content in bulk biomass of wild-harvested and cultivated edible seaweeds: inherent variations determine species-specific daily allowable consumption. Food Chemistry 254:333-339. https://doi.org/10.1016/j.foodchem.2018.02.024

de la Moneda A, et al. (2019) Variability and potential of seaweeds as ingredients of ruminant diets: an in vitro study. Animals 9:851. https://doi.org/10.3390/ani9100851

Ramin M, et al. (2019) In vitro evaluation of utilisable crude protein and methane production for a diet in which grass silage was replaced by different levels and fractions of extracted seaweed proteins. Animal Feed Science and Technology 255: 114225. https://doi.org/10.1016/j.anifeedsci.2019.114225

Stévant P, et al. (2018) Effects of drying on the nutrient content and physico-chemical and sensory characteristics of the edible kelp Saccharina latissima. Journal of Applied Phycology 30:2587-2599. https://doi.org/10.1007/s10811-018-1451-0

Gaillard C, et al. (2018) Amino acid profiles of nine seaweed species and their in situ degradability in dairy cows. Animal Feed Science and Technology 241:210-222. https://doi.org/10.1016/j.anifeedsci.2018.05.003 

Bjarnadóttir M, etal. (2018) Palmaria palmata as an alternative protein source: Enzymatic protein extraction, amino acid composition, and nitrogen-to-protein conversion factor. Journal of Applied Phycology 30:2061-2070. https://doi.org/10.1007/s10811-017-1351-8

Molina-Alcaide E, et al. (2017) In vitro ruminal fermentation and methane production of different seaweed species. Animal Feed Science and Technology 228:1-12. https://doi.org/10.1016/j.anifeedsci.2017.03.012

Tayyab U, et al. (2016) Ruminal and intestinal degradability of various seaweed species measured in situ in dairy cows. Animal Feed Science and Technology 213: 44-54. https://doi.org/10.1016/j.anifeedsci.2016.01.003

Author Response

First of all, we are highly thankful to the learned reviewer for sparing time to review our paper during this ongoing deadly pandemic. Then, we are also highly thankful for all the improvement and the input received.               

Q1. Whole manuscript, the English needs improvement. Manuscript for review has several typographical errors.

A1. The linguistic revision and the topographical errors have been fixed.

Q2. Change Title. Suggestion- Chemistry of tropical eucheumatoids: potentials for food and feed applications.

A2. The title has been changed as indicated by the reviewer.

Q3. Abstract needs improvement. Not comprehensive enough in describing the extent of chemical characterization done.

A3. The required improvements have been done and chemical characterization has been included.

Q4. Line 28: Why K. alvarezii SHOULD be used in animal nutrition? Is this imperative?

A4. The authors have corrected: K. alvarezii could be used as potential additive in feed because its chemical and nutritional features (see line31).

Materials and method: require details.

Q5. Are the seaweed samples wild or cultivated? How were the different species recognized from each other? Based on morphology (be categorical. Include pictures of the seaweed as a Figure)? They could look the same among cultivated/farmed specimen. See Roleda et al. 2021 Algae 

A5. The seaweeds are cultivated, as indicated in the abstract, materials and methods, results and discussion.

In Materials and Methods we highlighted to see Roleda et al. 2021 for morphological characteristics : In the next manunuscript we ‘ll perform barcoding and morphological identification.

   Q6. How was the biomass sampling/collection done?

   A6.We reported in the revised manuscript how was done sampling/ collection (see lines 119-141).

Q7. What does n=6 pools/year mean?

A7. N= 6 pools corresponds to six baskets used as reported in Materials and Methods. All details are reported in this section (see lines 119-131).

Q8. What is a “pool”?

A8. It’s the content in a single basket.

Q9.How much sample i.e. wet weight per replicate was collected. How were they dried for eventual chemical assays?

A9. In the revised manuscript the used quantities are reported in each essays. The replicas are reported  in Materials and Methods (see lines 136-138).

Q10. Does this means that both species were collected 2 times (2018 and 2019)?

A10. The revised manuscript reported cleary that the two Kappaphycus species were collected in 2018 and 2019.

Q11. Does all data represent mean of 2 years?

A11. Data represent mean of 2 years (see Materials and Methods line 124).

Q12.This has implication on the statistical analysis. Comparison (1-way ANOVA) was only between species.

A12. We apologize for having mistakenly sent a version without the statistical analysis .

 Q13. Physico-chemical parameters of sea water. How was sampling done? Replications? Snapshot measurement? What is their relevance to the chemistry of the seaweed tissue/biomass?

A13 The  table 1 has been removed and the temperature, which is a determining factor, has been added in the text. The reason for this elimination is due to the fact that the sampling of the algae is not followed when the water conditions change. The sampling and replicates were performed following reference 28. The measurement was snapshot. We cannot know their relevance because the sampling did not take place.

Q14 Discussion: compare and contrast the chemistry of the 2 Kappaphycus species with other seaweeds with known food and feed applications (see recommended literatures below).

A14. According to the suggestion of the reviewer, the discussion has been improved thanks to all indicated references. They have also added relevant informations concerning the potential use of the seaweeds in different animal species.

Reviewer 2 Report

This is an interesting paper dealing with the chemical characterisation and the antioxidant potential of two wild species of Kapphaphycus from Malaysia.

Although I acknowledge the effort of the authors to produce a structured relevant scientific paper, the information provided is scarce and barely discussed.

The abstract needs to be reviewed in order to be quickly understandable the analysis performed, the results obtained and the relevance of such results.

The introduction needs further information regarding the theme.

The methodology is presented with accuracy and follow standard methods.

The results, as mentioned, are scarce. Namely, the antioxidant capacity is only evaluated through ABTS when other common methods should have been used to perform a comprehensive analysis, namely DPPH, TPC, FRAP.

The discussion is very poor and must be clearly improved. Because the data are scarce, these must be thorough analysed, comparing with other species and populations, with environmental parameters and the potential of the biomass as feed in different animal groups.

Also, this doesn’t seem to be the final version of the document, for there are many mistakes throughout the text that need t be corrected.

Thus, I have some doubts regarding the publication of this documents. I recommend the authors to make major changes on the document. 

Specific comments are made below to improve the document:

The abstract needs reviewing. A sentence introducing the subject is advised. The main analysis performed are not referred, but then the results refer them: heavy metals, protein, ... and these are important analysis for the evaluation of the species as feed.

Line 20: correct thetotal – the total;

correct: capacityin – capacity in

Line 23:correct:  animalconsumption – animal consumption

Line 26: substitute samples by species

Line 28: correct: concludedthat – concluded that

Introduction:

Some information is missing from the introduction, namely the common chemical composition of seaweeds, to compare later with the two species analysed in the study.

Also, there are plenty of papers related to the use of Kappaphyccus in feed. Some of these should also be referred to in the introduction.  Eg.

https://doi.org/10.1016/j.anifeedsci.2020.114786

https://doi.org/10.1016/0044-8486(96)01282-3

https://doi.org/10.1002/jsfa.10708

https://doi.org/10.1016/S0044-8486(96)01393-2

In the final paragraph, either the authors state all the chemical analysis performed or none. In this context, protein, fat and fibre are mandatory to assess the potential for the feed industry.

Line 49: correct: Kappaphycusalvarezii – Kappaphycus alvarezii

Line 49. The first time the scientific name of a species is mentioned it should indicate its full name:

Kappaphycus alvarezii (Doty) L.M.Liao 1996

Kappaphycus striatus (F.Schmitz) L.M.Liao 1996

Then, you can use the “short” version.

Line 68: I’m not sure in other parts of the globe, but in Europe the toxicity of seaweeds (amount of heavy metals) is regulated by the EU regulation 1881/2006. This may be specifically refered, because the authors make references to it in the discussion.

Line 70: delete as.

Line 71: add an “s” to animal, add a full stop.

Line 75: correct: seaweeds,asall to seaweeds, as all – I desagree with the sentence. It is true that all organisms have mechanisms to deal with ROS, but seaweeds (among others) have very efficient special secondary metabolites to dead with ROS.

Line 76: delete Therefore, [16].

Line 86 – seaaweed is plural in this context, I believe.

Line 89 – correct wasto – was to

Line 90 - correct carrageenansand – carrageenans and

Line 92 – correct capacityin – capacity in

Methods:

Line 104 – in our lab, we found out that washing with seawater is better than milliQ water, due to the possible diffusion of compounds out of the seaweeds. Also, when analysing bioactivities, we dry the biomass at 25ºC (30ºC) which is room temperature, in order to preserve the chemical structure and thus the bioactivity of the metabolites.

Line 181 - correct capacitywere – capacity were

correct byMann – by Mann

Results:

Line 188 - correct alvareziishowed – alvarezii showed

line 198 – correct striatuswere – striatus were

line 212 – correct althoughκ – although κ

line 225 – correct analysedbyMannWhitney

Figure 2 – the caption is not very clear; please rephrase the differences between c) and d). Also, there are not differences in K. striatus between soluble and fat soluble? Analysing the graphic, I find that hard to believe.

Discussion

Line 236 – correct towardshealth

Line 240 – consider substituting “help” by “benefit”

Line 241 – discuss the nutritional profile: protein, fat and fibre, … and the quality of the biomass as potential feed.

Line 242 - correct demonstratedthere

Line 245 – correct Inthe light; correct haveevaluated

Line 249 – correct whichKappaphycus

Line 251 – as stated abone, the names of the authors (and dates) should be stated the first time you mention the species, and only once. The names the authors used are incomplete – see above.

Line 261 – rephrase “sort of” into a more formal sentence.

Line 262 – are these results in accordance with other authors? Do similar species present similar/different results?

Line 266 – How were they affected by environmental factors? Can you relate the chemical properties of the biomass with the parameters referred to in table 1? Otherwise, why do the authors show the data of this table if it is not discussed?

Line 270 to 273 – this sentence has nothing to do with bioavailability – bioavailability for dietary supplements can be defined as the proportion of the administered substance capable of being absorbed and available for use or storage (doi:10.1093/jn/131.4.1349S). Although it is affected by the concentration, it is also influenced by solubility, interaction between nutrients, circadian differences, gastric properties,  metabolic differences, disease, and many other parameters.

Thus, all references to bioavailability should be rephrase accordingly.

Line 274 – correct - studieswere

Line 275 – correct - asthe

Line 276 – correct - sincetrace

Line 274 to 283 – these results should be further discussed. Are they interesting for Fish? Piggery? Poultry? (…) Each group has specific requirements, and this should be discussed, either the chemical organic analysis, or the minerals. Can the authors relate the results obtained with the actual needs of each group of animal’s feed?

Line 302 – correct Bidongsuggested

Line 305 correct – striatusfraction

Line 308 – there is no discussion regarding carrageenans. Are these results interesting? Comparing with other species/populations, can this carrageenans be used in the food industry?

Line 309 – regarding the antioxidant capacity, using only one method is truly limiting for each method search for a group of antioxidant molecules, meaning that different methods allow to search for diverse groups of molecules. Even though, the results are interesting but not discussed with the findings from other authors for the same species and similar species, other environments, farmed species, ...

Again, the impact of the environmental factors may be discussed, namely the temperature.

Author Response

First of all, we are highly thankful to the learned reviewer for sparing time to review our paper during this ongoing deadly pandemic. Then, we are also highly thankful for all the improvement and the input received.

Q1 The abstract needs reviewing. A sentence introducing the subject is advised. The main analysis performed are not referred, but then the results refer them: heavy metals, protein, ... and these are important analysis for the evaluation of the species as feed.

A1. The abstract has been improved; a sentence has been introduced the subject and the performed analysis were referred.

Q2. Line 20: correct thetotal – the total;correct: capacityin – capacity in

Line 23:correct:  animalconsumption – animal consumption

Line 26: substitute samples by species

Line 28: correct: concludedthat – concluded that

A2. We apologize for having mistakenly sent a version with formatting problems related to different versions of word.

Q3. Introduction:

Some information is missing from the introduction, namely the common chemical composition of seaweeds, to compare later with the two species analysed in the study.

Also, there are plenty of papers related to the use of Kappaphyccus in feed. Some of these should also be referred to in the introduction. Eg.

https://doi.org/10.1016/j.anifeedsci.2020.114786

https://doi.org/10.1016/0044-8486(96)01282-3

https://doi.org/10.1002/jsfa.10708

https://doi.org/10.1016/S0044-8486(96)01393-2

In the final paragraph, either the authors state all the chemical analysis performed or none. In this context, protein, fat and fibre are mandatory to assess the potential for the feed industry.

A3. The requests of reviewer have been fulfilled using indications about the suggested literature

Q4. Line 49: correct: Kappaphycusalvarezii – Kappaphycus alvarezii

Line 49. The first time the scientific name of a species is mentioned it should indicate its full name:

Kappaphycus alvarezii (Doty) L.M.Liao 1996

Kappaphycus striatus (F.Schmitz) L.M.Liao 1996

Then, you can use the “short” version.

A4. Done. Line 54

Q5. Line 68: I’m not sure in other parts of the globe, but in Europe the toxicity of seaweeds (amount of heavy metals) is regulated by the EU regulation 1881/2006. This may be specifically refered, because the authors make references to it in the discussion.

A5. The “Heavy metals” regulation of EU has been reported (line81)

Q6. Line 70: delete as.

A6. Done

Q7: Line 71: add an “s” to animal, add a full stop.

A7. Done

Q8. Line 75: correct: seaweeds,asall to seaweeds, as all – I desagree with the sentence. It is true that all organisms have mechanisms to deal with ROS, but seaweeds (among others) have very efficient special secondary metabolites to dead with ROS.

A8. The corrections have been done. At line 90, the authors refer about secondary metabolites

Q9.

a-Line 76: delete Therefore,[16].

b-Line 86 – seaaweed is plural in this context, I believe.

c-Line 89 – correct wasto – was to

d-Line 90 - correct carrageenansand – carrageenans and

e-Line 92 – correct capacityin – capacity in

A9. a, c-e Done; b- done, see line98

Discussion

Q10.

Line 236 – correct towardshealth

Line 240 – consider substituting “help” by “benefit”

Line 241 – discuss the nutritional profile: protein, fat and fibre, … and the quality of the biomass as potential feed.

Line 242 - correct demonstratedthere

Line 245 – correct Inthe light; correct haveevaluated

Line 249 – correct which Kappaphycus

A10. done

Q11. Line 251 – as stated abone, the names of the authors (and dates) should be stated the first time you mention the species, and only once. The names the authors used are incomplete – see above.

A11.  The names and dates are reported at line 54.

Q12. Line 261 – rephrase “sort of” into a more formal sentence.

A12.  “Sort” has been has been replaced with “double”

Q13. Line 262 – are these results in accordance with other authors? Do similar species present similar/different results?

A13. Other results in similar species are reported in the edited text (see from line 377 to 434).

Q14. Line 266 – How were they affected by environmental factors? Can you relate the chemical properties of the biomass with the parameters referred to in table 1? Otherwise, why do the authors show the data of this table if it is not discussed?

A14. Thank you for your observation. The environmental factors of the harvest site seems able to affect the chemical characteristics of the seaweeds. However in the table 1 we report only the characteristics of our sampling site, but unfortunately we did not compare the characteristics of the seaweeds harvest in different sampling conditions. So, we deleted Tab. 1.

Q15. Line 270 to 273 – this sentence has nothing to do with bioavailability – bioavailability for dietary supplements can be defined as the proportion of the administered substance capable of being absorbed and available for use or storage (doi:10.1093/jn/131.4.1349S). Although it is affected by the concentration, it is also influenced by solubility, interaction between nutrients, circadian differences, gastric properties,  metabolic differences, disease, and many other parameters.

Thus, all references to bioavailability should be rephrase accordingly.

A15.  The concept of  bioavailability has been rectified

As you can see at lines (343-347)

Q16.

Line 274 – correct - studieswere

Line 275 – correct - asthe

Line 276 – correct – sincetrace

A16. Done

Q17. Line 274 to 283 – these results should be further discussed. Are they interesting for Fish? Piggery? Poultry? (…) Each group has specific requirements, and this should be discussed, either the chemical organic analysis, or the minerals. Can the authors relate the results obtained with the actual needs of each group of animal’s feed?

A17. All the literature cited from line 403 to line 432 refers to cows. According to the suggestion of the reviewer, the discussion has been improved.

Q18.

Line 302 – correct Bidongsuggested

Line 305 correct – striatusfraction

A18. Done

Q19. Line 308 – there is no discussion regarding carrageenans. Are these results interesting? Comparing with other species/populations, can this carrageenans be used in the food industry?

A19. The importance of carrageenan and variations in other algae of other species and populations have been discussed ( from line 435 to 445).

Q20. Line 309 – regarding the antioxidant capacity, using only one method is truly limiting for each method search for a group of antioxidant molecules, meaning that different methods allow to search for diverse groups of molecules. Even though, the results are interesting but not discussed with the findings from other authors for the same species and similar species, other environments, farmed species, ...

Again, the impact of the environmental factors may be discussed, namely the temperature.

A20. Total soluble and fat soluble antioxidant capacities have also measured by the FRAP and DPPH essays. Since data on the antioxidant capacity of seaweeds depend on their growth conditions and extraction method, it has been not possible to report any comparison as required

Round 2

Reviewer 1 Report

English language and M&M still require significant improvement. Please find attached commented pdf.

The corresponding author is requested to copy and paste the comments in the annotated pdf and address each comment succinctly, and refer to the corresponding changes instituted in the revised manuscript.

Authors are request to submit both an annotated and a clean versions of the manuscript.

Author Response

Q1. English language and M&M still require significant improvement. Please find attached commented pdf.

A.1. Significant improvement has been obtained after the English revision of a native English-speaking colleague.Thanks for your suggestion. Please see attached pdf

Q2.The corresponding author is requested to copy and paste the comments in the annotated pdf and address each comment succinctly, and refer to the corresponding changes instituted in the revised manuscript.

A.2. The comments in the annotated pdf have been done as requested and each comment succinctly addressed, and the corresponding changes instituted were referred in the revised manuscript. 

 Q2a. Wild? Or cultivars from a seaweed farm? 

A.2a.The words“in the wild offshore” have been added.

 Q2b1. This paragraph is totally problematic. The English is still not good enough. Needs clarity.  

A2b1. Significant improvement has been obtained after the English revision of a native English-speaking colleague. 

 Q2b2. After preliminary species identification, was there any confirmation done? Not easy to discriminate K. alvarezii from K. striatus specially among cultivated strains. How sure are you that what were studied were distinct species and not different varieties/cultivars/strains of the same species? The same species could exhibit a wide range of morphologies.  

A2b2. Besides identification from direct knowledge by the oceanic engineer Olanrewaju,  verification was also confirmed as described in the works indicated in references 26 and reported by Roleda et al. 2021 (27).

 Q2b3. What was done in 2018 and 2019? The same in situ growth experiments? Why done in 2 separate years? 

A2b3.  The growth are related to many approaches of our research and we use to repeat all the experiments at least for two years before writing our manuscripts. 

To insure repeatability and to account for any large variation in environmental conditions, we repeat all of our experiments for at least two years before reporting our results.

------------------------------------------------------------------------------- 

Q2b4. K. alvarezii and K. striatus growth experiments done simultaneously in each year? Why measure growth only for 2 weeks? Typical growing/farming of eucheumatoids takes 45-50 days.

A2b4. The growth of the seaweeds was followed for 50 days in both years preceding our experiments. Maximum growth was seen after 14 days. Therefore we considered it appropriate to check the chemical and physical conditions of the water and samples (DRG) and to collect the seaweeds after 2 weeks.

 Q2b5a. Provide the initial weight (mean and range)for all the baskets instead of g per liter stocking density.

A2b5a. Mean±SD Initial weights for each set of 6 baskets of K. alvarezii and K. striatus were 22.25 ±0.3g and25.2±0.5g respectively and included in the text as requested

Q2b5b. How many baskets per species (= replication)?

A2b5b. A total of 6 baskets per specie sper year. This was clarified in the text.

Q2b6a. After 2 weeks, what was the final weight? Provide mean and ranges for each species on top of the reported growth rates. Report the mean growth rates during the 2-week experimental period, and not the maximum growth rates.  

A2b6a. After two weeks final weight of K alvarezii and K. striatus was 3173±120 g and 3239±103 g respectively, because average daily growth rates (DGRs) were 5.30%·day-1 and  5.45%·day-1 with corresponding biomass of 225± 18 g  and 231± 15 g.

Q2b6b. What is biweekly? 2x a week? Better state "biomass was measured every x days and mean growth rate was measured using the xxx equation". 

A2b6b. Biomass was measured 2x a week and not every day, because it was detected every days two years before our experiments. The mean growth rate was followed by the equation: DGR (%)= In (Wf /Wo)/ t x 100 where Wf is the final fresh weight (g) at t day, Wo is the initial fresh weight (g), and t is the number of culture days. In particular we have deleted from text the sentence “….biweekly….” to avoid confusion to the readers.

  Q2b7.  It is important to know the final weight after 2 weeks because after drying (approximately 80% water), was there enough dry biomass per replicate to measure all the chemical variables? 

A2b7. The final weigth was reported in paragraph 2.1. “Biological material” in Materials and Methods. The quantity of obtained material was greater than that necessary to carry out all the required determinations.

 Q2b8. Provide photo of the cultivation set-up and the "basket", and the fresh seaweed (visible gross morphology). 

A2b8. We provided in Figure 1 a photo of the cultivation set-up and "basket system and the fresh seaweed species.

Q2b9.Check English!

A2b9. Done

 Q2b10 a.Triplicate analyses for all chemical parameters for each species in 2018 and 2019 harvested biomass? Then how was the data treated and analyzed?  

A2b10a. Results relating to determination of chemical characteristics, mineral and heavy metal composition and biomass and total soluble and fat soluble antioxidant capacities obtained in 2018 and 2019 in K. alvarezii and K. striatus were very similar, thus they were reported as mean ± standard deviation (SD).

Chemical characteristics, mineral and heavy metal composition and biomass results were analyzed using one-way ANOVA. Root mean square error (RMSE ) was reported. Differences were considered significant when P< 0.01  and P< 0.058.

 Q2b10b. Why averaged the observations for the 2 years? This was stated in the response to reviewer's comment but not specified in the M&M. WHY? Why not present separately as chemistry for 2018 and 2019. 

A2b10b. The chemical and biochemical data represent anmean of 36 values obtained in 2018 and 36 values in 2019 for each species.The values of 2018 and 2019 were very similar, therefore we considered not appropriate to report them separately .

 Q2b11. Not easy to discrimate among fresh/live samples.Much more among dried samples. Farmed strain could easily be mixed up by farmers. 

A2b11. Although it is not easy to discriminate among fresh /live samples and even moresoamong dried  samples, but not so for the farmers,who have an  intimate working knowledge of the differences between the two species. For the scientist, we suggest followingthe  morphological characterization as described in Doty [26] reported in Roleda et al. [27].

In particular K. striatus shows variable thalli forms, from “Elkhorn” shape with secondary branching near apices deflects the straight direction of the primary branch forming an antler like pattern to a form with dichotomous branching of primary and secondary branches.

 We believe that experienced farmers could hardly confuse the two species and,in addition, for our experiments they respected and maintained the separation between one basket content and the other during drying. The separation allowed us to collect two different aliquots of seaweeds from each basket.

Q2b12. In the text below, please specify the mount of dried samples required for each analysis.

A2b12. All analyzes were performed on dried samples. For the chemical determinations, two aliquots of 0.5 g were taken from each basket; for the analysis on antioxidants, two aliquots of 0.3g; for the analysis on carrageenans, two  aliquots of 10 g.

 Q2b13 How were the 2018 and 2019 data treated and analyzed? 

A2b13. Data relating to determination of chemical characteristics, mineral and heavy metal composition and biomass and total soluble and fat soluble antioxidant capacities obtained in 2018 and 2019 in K. alvareziiand K. striatus were very similar, thus they were reported as mean ± standard deviation (SD).

Chemical characteristics, mineral and heavy metal composition and biomass results were analyzed using one-way ANOVA. Root mean square error (RMSE ) was reported. Differences were considered significant when P< 0.01 (A, B) and P< 0.058 (a, b).

Total soluble and fat soluble antioxidant capacities measured by FRAP, DPPH and ABTS test were analyzed by the Mann-Whitney U test and they are shown as mean ± SD. Differences were considered significant when P<0.05.

 Q2b14a.Are these the average for 2018 (n=) and 2019 (n=) samples? 

A2b14a. Two aliquots among the specimens in 6 baskets represent for us 12 samples. The chemical analysis has been carried out in triplicate on 12 samples. Therefore 36 analysis has been performed in one year for each species.

 Q2b14 b.What is the total n for 2 years?

A2b14b.The total analysis done in two years have been 72

 Q2b14 c.Whereare the SD or SE values?

A2b14c. In table 1, the SD are reported as RMSE: root mean square error.

 Q2b15. How was this computed? Almost 80% of fresh (wet) seaweed is water.

A2b15. We used dried seaweeds.

Q2b16.What is this?

A2b16. This is the continuation of the previous sentence the histogram 4.

Q2b17a. Be clear in the figure caption.

A2b17a. The figure caption was has been clarified following the suggested indications.

Q2b17b.Do these bar graphs represent the mean for 2018 and 2019?

A2b17b. Yes

Q2b17c. What is the total n?

A2b17c. The biochemical data represent a mean of 36 values obtained analyzing two aliquots collected from each of the 6 baskets and because the antioxidant determinations were conducted in triplicate for each of the 12 aliquots in each year, 2018-2019  for each species. (thus 6 baskets  x 2 aliquots x 3 replicated analyses =36)

 Q2b17d. Are the error bars SD or SE?

A2b17d.In the bar charts4 and 5 the error bars are ±SD.

Q2b17b1. Do these bar graphs represent the mean for 2018 and 2019?

A2b17b1. Yes

 Q2b17c1. What is the total n?

A2b17c1. The biochemical data represent a mean of 36 values obtained analyzing two aliquots collected from each of the 6 baskets and because the antioxidant determinations were conducted in triplicate for each of the 12 aliquots in each year, 2018-2019  for each species. (thus 6 baskets  x 2 aliquots x 3 replicated analyses =36)

 Q2b17d1. Are the error bars SD or SE?

A2b17d1. In the bar charts 4 and 5 the error bars are ±SD

Q2b17e. HOW COME n=5 here? Doesn't make sense when I have the impression that for each year i.e., 2018 and 2019, n=3. Then total n should not be 6?

A2b17e. Apologize for mistake, n=36 ( as in A2b17c1)

 Q2b17e. HOW COME n=5 here? Doesn't make sense when I have the impression that for each year i.e., 2018 and 2019, n=3. Then total n should not be 6? 

A2b17e. Apologize for mistake, “n=36, see A2b17c1 

 Q2b18.Did a random check. These references [17, 21] don't seem to correspond to "seaweeds' substantial benefits for human health". Duckweed is not seaweed!  

Authors, please double check your references. 

  1. Abdel-Gawad, F.K.; Khalil, W.K.B.; Bassem, S.M.; Kumar, V.; Parisi, C.; Inglese, S.; Temraz, T.A.; Nassar, H.F.; Guerriero, G. The Duckweed, Lemna minor Modulates Heavy Metal-Induced Oxidative Stress in the Nile Tilapia, Oreochromis niloticus. Water 2020, 12, 2983, doi:10.3390/w12112983. 

  1. Mizpal, A.; Rosazman, H.; Suhaimi, M.; Ahmad Tarmizi, A. Projek estet mini rumpai laut dan penglibatan komuniti nelayan di daerah Semporna, Sabah. J. Borneo Transform. Stud. 2015, 1, 122–138. 

A2b18. We have reported the relevance of duckweed [17] and seaweed[ 21] for human health .

 Q2b19a .What do you mean? Kappaphycus alvarezii and Kappaphycus striatus both belong to one genus i.e., Kappaphycus.  

A2b19a. From our competence in phylogenetics, Kappaphycusis the genus and striatus and alvarezii are the names of the two different species 

Q2b19b.You mean which Kappaphycus species? How sure are you that they are 2 species? Without molecular confirmation, maybe they were just 2 different varieties/ cultivars/ strains of the same species? 

A2b19b. The primers successfully used for Kappaphycus discrimination were published in 2021. Prof. Guerriero is our expert for barcoding,she will perform the experiment after receiving permission from our department. Morphological identification was done by Prof. Olanrewaju, the author who permits our seaweed studies. Furthermore we used the literature you have suggested as reference [26] reported in Roleda et al. [27]. 

 Q2b20. What do you mean? They are distinct species. Not varieties. 

A2b20.  This was due to misinterpretation by the translator. Text was corrected to say “species”.

Q2b21. How did you know that there was a decrease in growth rates after 14 days when the experimental set-up was supposed to have been harvested on day 14 for chemical analyses? 

A2b21. This was based on previous growth experiments which were performed before our project. 

Q2b22. Check English. "ubiquitous" ??? 

A2b22. Edited. 

 Q2b23. Check English of the whole manuscript! 

A2b23. Done by a native English-speaking colleague. 

 Q2b 24. What do you mean? Check English! 

A2b24. The sentence has been reworded deleting “along with their not animal nature”.

-------------------------------- 

Q3. Authors are request to submit both an annotated and a clean versions of the manuscript. 

A.3. Done as requested?

Q1. English language and M&M still require significant improvement. Please find attached commented pdf.

A.1. Significant improvement has been obtained after the English revision of a native English-speaking colleague.Thanks for your suggestion.

Q2.The corresponding author is requestedto copy and paste the comments in the annotated pdf and address each comment succinctly, and refer to the corresponding changes instituted in the revised manuscript.

A.2. The comments in the annotated pdf have been done as requested and each comment succinctly addressed, and the corresponding changes instituted were referred in the revised manuscript. 

 Q2a. Wild? Or cultivars from a seaweed farm? 

A.2a.The words“in the wild offshore” have been added.

 Q2b1. This paragraph is totally problematic. The English is still not good enough. Needs clarity.  

A2b1. Significant improvement has been obtained after the English revision of a native English-speaking colleague. 

 Q2b2.After preliminary species identification, was there any confirmation done? Not easy to discriminate K. alvarezii from K. striatusspeciallyamong cultivated strains. How sure are you that what were studiedwere distinct species and not different varieties/cultivars/strains of the same species? The same species could exhibit a wide range of morphologies.  

A2b2. Besides identification from direct knowledge by the oceanic engineer Olanrewaju,  verification was also confirmed as described in the works indicated in references 26 and reported byRoleda et al. 2021 (27).

 Q2b3. What was done in 2018 and 2019? The same in situ growth experiments? Why done in 2 separate years? 

A2b3.  The growth are related to many approaches of our research and we use to repeat all the experiments at least for two years before writing our manuscripts. 

To insure repeatability and to account foranylarge variation in environmental conditions, we repeat all of our experiments for at least two years before reporting our results.

------------------------------------------------------------------------------- 

Q2b4.K. alvarezii and K. striatusgrowth experiments done simultaneously in each year? Why measure growth only for 2 weeks? Typical growing/farming of eucheumatoidstakes 45-50 days.

A2b4. The growth of the seaweeds was followed for 50 days in both years preceding our experiments. Maximum growth was seen after 14 days. Therefore we considered it appropriate to check the chemical and physical conditions of the water and samples (DRG) and to collect the seaweeds after 2 weeks.

 Q2b5a.Provide the initial weight (mean and range)for all the baskets instead of g per liter stocking density.

A2b5a.Mean±SD Initial weights for each set of 6 baskets of K.alvareziiandK. striatuswere 22.25 ±0.03g and25±0.5g respectively and included in the text as requested

Q2b5b.How many baskets per species (= replication)?

A2b5b.A total of 6 baskets per speciesper year. This was clarified in the text.

Q2b6a.After 2 weeks, what was the final weight? Provide mean and ranges for each species on top of the reported growth rates. Report the mean growth rates during the 2-week experimental period, and not the maximum growth rates.  

A2b6a.After two weeks final weight of K alvarezii and K. striatuswas3173±120 g and 3239±103 respectively, because average daily growth rates (DGRs) were5.30%·day-1 and  5.45%·day-1 with corresponding biomass of 225± 18 g  and 231± 15 g.

Q2b6b. What is biweekly? 2x a week? Better state "biomass was measured every x days and mean growth rate was measured using the xxx equation". 

A2b6b.Biomass was measured 2x a week and not every day, because it was detected every days two years before our experiments. The mean growth rate was followed by the equation: DGR (%)= In (Wf /Wo)/ t x 100 where Wf is the final fresh weight (g) at t day, Wo is the initial fresh weight (g), and t is the number of culture days. In particular we have deleted from text the sentence “….biweekly….” to avoid confusion to the readers.

 Q2b7. It is important to know the final weight after 2 weeks because after drying (approximately 80% water), was there enough dry biomass per replicate to measure all the chemical variables? 

A2b7.The final weigthwas reported in paragraph 2.1. “Biological material” in Materials and Methods. The quantity of obtained material was greater than that necessary to carry out all the required determinations.

 Q2b8.Provide photo of the cultivation set-up and the "basket", and the fresh seaweed (visible gross morphology). 

A2b8. We provided photo of the cultivation set-up and "basket system and the fresh seaweed (visible with gross morphology.xxxxx

Q2b9.Check English!

A2b9. Done

Q2b10 a.Triplicate analyses for all chemical parameters for each species in 2018 and 2019 harvested biomass? Then how was the data treated and analyzed?  

A2b10a.Results relating to determination of chemical characteristics, mineral and heavy metal composition and biomass and total soluble and fat soluble antioxidant capacities obtained in 2018 and 2019 in K. alvarezii and K. striatus were very similar, thus they were reported as mean ± standard deviation (SD).

Chemical characteristics, mineral and heavy metal composition and biomass results were analyzed using one-way ANOVA. Root mean square error (RMSE ) was reported. Differences were considered significant when P< 0.01 (A, B) and P< 0.058 (a, b).

Q2b10b.Why averaged the observations for the 2 years? This was stated in the response to reviewer's comment but not specified in the M&M. WHY? Why not present separately as chemistry for 2018 and 2019. 

A2b10b.The chemical and biochemical data represent anmean of 36 values obtained in 2018 and 36 values in 2019 for each species.The values of 2018 and 2019 were very similar, therefore we considered not appropriate to report them separately .

Q2b11.Not easy to discrimate among fresh/live samples.Much more among dried samples. Farmed strain could easily be mixed up by farmers. 

A2b11. Although it is not easy to discriminate among fresh /live samples and even moresoamong dried  samples, but not so for the farmers,who have an  intimate working knowledge of the differences between the two species. For the scientist, we suggest followingthe  morphological characterization as described in Doty [26] reported in Roleda et al. [27].

In particular K. striatus shows variable thalli forms, from “Elkhorn” shape with secondary branching near apices deflects the straight direction of the primary branch forming an antler like pattern to a form with dichotomous branching of primary and secondary branches.

 We believe that experienced farmers could hardly confuse the two species and,in addition, for our experiments they respected and maintained the separation between one basket content and the other during drying. The separation allowed us to collect two different aliquots of seaweeds from each basket.

Q2b12. In the text below, please specify the mount of dried samples required for each analysis.

A2b12. All analyzes were performed on dried samples. For the chemical determinations, two aliquots of 0.5 g were taken from each basket; for the analysis on antioxidants, two aliquots of 0.3g; for the analysis on carrageenans, two  aliquots of 10 g.

Q2b13 How were the 2018 and 2019 data treated and analyzed? 

A2b13.Data relating to determination of chemical characteristics, mineral and heavy metal composition and biomass and total soluble and fat soluble antioxidant capacities obtained in 2018 and 2019 in K. alvareziiand K. striatus were very similar, thus they were reported as mean ± standard deviation (SD).

Chemical characteristics, mineral and heavy metal composition and biomass results were analyzed using one-way ANOVA. Root mean square error (RMSE ) was reported. Differences were considered significant when P< 0.01 (A, B) and P< 0.058 (a, b).

Total soluble and fat soluble antioxidant capacities measured by FRAP, DPPH and ABTS test were analyzed by the Mann-Whitney U test and they are shown as mean ± SD. Differences were considered significant when P<0.05.

Q2b14a.Are these the average for 2018 (n=) and 2019 (n=) samples? 

A2b14a. Two aliquots among the specimens in 6 baskets represent for us 12 samples. The chemical analysis has been carried out in triplicate on 12 samples. Therefore 36 analysis has been performed in one year for each species.

Q2b14 b.What is the total n for 2 years?

A2b14b.The total analysis done in two years have been 72

Q2b14 c.Whereare the SD or SE values?

A2b14c.In table 1, the SD are reported as RMSE: root mean square error.

Q2b15.How was this computed? Almost 80% of fresh (wet) seaweed is water.

A2b15. We used dried seaweeds.

Q2b16.What is this?

A2b16.This is the continuation of the previous sentence the histogram 4.

Q2b17a. Be clear in the figure caption.

A2b17a. The figure caption was has been clarified following the suggested indications.

Q2b17b.Do these bar graphs represent the mean for 2018 and 2019?

A2b17b.Yes

Q2b17c.What is the total n?

A2b17c. The biochemical data represent a mean of 36 values obtained analyzing two aliquots collected from each of the 6 baskets and because the antioxidant determinations were conducted in triplicate for each of the 12 aliquots in each year, 2018-2019  for each species. (thus 6 baskets  x 2 aliquots x 3 replicated analyses =36)

Q2b17d.Are the error bars SD or SE?

A2b17d.In the bar charts4 and 5 the error bars are ±SD.

Q2b17b1.Do these bar graphs represent the mean for 2018 and 2019?

A2b17b1.Yes

Q2b17c1.What is the total n?

A2b17c1 The biochemical data represent a mean of 36 values obtained analyzing two aliquots collected from each of the 6 baskets and because the antioxidant determinations were conducted in triplicate for each of the 12 aliquots in each year, 2018-2019  for each species. (thus 6 baskets  x 2 aliquots x 3 replicated analyses =36)

Q2b17d1.Are the error bars SD or SE?

A2b17d1.In the bar charts4 and 5 the error bars are ±SD

Q2b17e.HOW COME n=5 here? Doesn't make sense when I have the impression that for each year i.e., 2018 and 2019, n=3. Then total n should not be 6?

A2b17e. Apologize for mistake, n=36 ( as in A2b17c1)

 Q2b17e.HOW COME n=5 here? Doesn't make sense when I have the impression that for each year i.e., 2018 and 2019, n=3. Then total n should not be 6? 

A2b17e. Apologize for mistake, “n=36, see A2b17c1 

 Q2b18.Did a random check. These references [17, 21] don't seem to correspond to "seaweeds' substantial benefits for human health". Duckweed is not seaweed!  

Authors, please double check your references. 

  1. Abdel-Gawad, F.K.; Khalil, W.K.B.; Bassem, S.M.; Kumar, V.; Parisi, C.; Inglese, S.; Temraz, T.A.; Nassar, H.F.; Guerriero, G. The Duckweed, Lemna minor Modulates Heavy Metal-Induced Oxidative Stress in the Nile Tilapia, Oreochromis niloticus. Water 2020, 12, 2983, doi:10.3390/w12112983. 

  1. Mizpal, A.; Rosazman, H.; Suhaimi, M.; Ahmad Tarmizi, A. Projek estet mini rumpai laut dan penglibatan komuniti nelayan di daerah Semporna, Sabah. J. Borneo Transform. Stud. 2015, 1, 122–138. 

A2b18. We have reported the relevance of duckweed [17] and seaweed[ 21] for human health .

 Q2b19a .What do you mean? Kappaphycus alvarezii and Kappaphycus striatus both belong to one genus i.e., Kappaphycus.  

A2b19a.From our competence in phylogenetics, Kappaphycusis the genus and striatus and alvarezii are the names of the two different species 

Q2b19b.You mean which Kappaphycus species? How sure are you that they are 2 species? Without molecular confirmation, maybe they were just 2 different varieties/ cultivars/ strains of the same species? 

A2b19b.The primers successfullyusedforKappaphycusdiscriminationwerepublished in 2021. Prof.Guerrierois our expert forbarcoding,shewillperform the experimentafter receiving permission from our department.Morphological identification was done by Prof. Olanrewaju, the author whopermits our seaweed studies. Furthermore we used the literature you have suggested as reference [26] reported in Roleda et al. [27]. 

Q2b20. What do you mean? They are distinct species. Not varieties. 

A2b20.  This was due to misinterpretation by the translator. Text was corrected to say “species”

Q2b21. How did you know that there was a decrease in growth rates after 14 days when the experimental set-up was supposed to have been harvested on day 14 for chemical analyses? 

A2b21. This was based on previous growth experiments which were performed before our project. 

Q2b22. Check English. "ubiquitous" ??? 

A2b22. Edited. 

Q2b23. Check English of the whole manuscript! 

A2b23. Done by a native English-speaking colleague. 

Q2b 24. What do you mean? Check English! 

A2b24. The sentence has been reworded deleting “along with their not animal nature”.

-------------------------------- 

Q3. Authors are request to submit both an annotated and a clean versions of the manuscript. 

A.3. Done as requested?

Reviewer 2 Report

I believe that the authors have succeeded in significantly improving the paper, addressing the issues that were previously highlighted in the document. They also responded very clearly to the questions that were raised.
New information has been added to the discussion, as requested, clearly improving it, regarding the chemical composition, the carrageenans and the antioxidant capacity of the species studied.  

Thus, the paper has been much improved

I, therefore, believe that it is in a position to be published with minor changes.

Line 131 – who banned the use of antibiotics? This sentence should be rephrased.

Table 1 – For Al : K.striatus 0.307, the “B” is above the line.- For As, the letters for significant differences are a and b, not A and B, for P=0.0343;

Line 383 – the authors started their trials with a high stocking density (5g/L). The decrease of growth during the second half of the experiment should be expected due to light shadowing and high seaweed density leading to competition. Maybe the authors could explain this better.

Line 400 – correct bio-availability to bioavailability

Line 443 – “Brown, Red and Green” should be written in lowercase letters.

Author Response

Q1. Line 131 – who banned the use of antibiotics? This sentence should be rephrased. 

A.1. The sentence was corrected

Q2. Table 1 – For Al : K.striatus 0.307, the “B” is above the line.- For As, the letters for significant differences are a and b, not A and B, for P=0.0343; 

A2 the table shows RMSE and P-value, therefore it has been remodeled without letters to avoid unnecessary additions

Q3. Line 383 – the authors started their trials with a high stocking density (5g/L). The decrease of growth during the second half of the experiment should be expected due to light shadowing and high seaweed density leading to competition. Maybe the authors could explain this better. 

A3. The authors have enriched the explanation.Thank you for your suggestion.

 Q4.Line 400 – correct bio-availability to bioavailability 

A.4. Done.

Q5.Line 443 – “Brown, Red and Green” should be written in lowercase letters. 

A.5. Done.